# A Novel Zinc Chelator, 1H10, Ameliorates Experimental Autoimmune Encephalomyelitis by Modulating Zinc Toxicity and AMPK Activation

**DOI:** 10.3390/ijms21093375

**Published:** 2020-05-10

**Authors:** Bo Young Choi, Jeong Hyun Jeong, Jae-Won Eom, Jae-Young Koh, Yang-Hee Kim, Sang Won Suh

**Affiliations:** 1Department of Physiology, Hallym University College of Medicine, Chuncheon 24252, Korea; bychoi@hallym.ac.kr (B.Y.C.); jd1422@hanmail.net (J.H.J.); 2Department of Molecular Biology, Sejong University, Seoul 05006, Korea; jweom90@hanmail.net (J.-W.E.); yhkim@sejong.ac.kr (Y.-H.K.); 3Department of Neurology, University of Ulsan College of Medicine, Seoul 138-736, Korea; jkko@amc.seoul.kr

**Keywords:** experimental autoimmune encephalomyelitis, zinc, AMPK, multiple sclerosis, microglia, BBB disruption, MMP-9

## Abstract

Previous studies in our lab revealed that chemical zinc chelation or zinc transporter 3 (*ZnT3*) gene deletion suppresses the clinical features and neuropathological changes associated with experimental autoimmune encephalomyelitis (EAE). In addition, although protective functions are well documented for AMP-activated protein kinase (AMPK), paradoxically, disease-promoting effects have also been demonstrated for this enzyme. Recent studies have demonstrated that AMPK contributes to zinc-induced neurotoxicity and that 1H10, an inhibitor of AMPK, reduces zinc-induced neuronal death and protects against oxidative stress, excitotoxicity, and apoptosis. Here, we sought to evaluate the therapeutic efficacy of 1H10 against myelin oligodendrocyte glycoprotein 35-55-induced EAE. 1H10 (5 μg/kg) was intraperitoneally injected once per day for the entire experimental course. Histological evaluation was performed three weeks after the initial immunization. We found that 1H10 profoundly reduced the severity of the induced EAE and that there was a remarkable suppression of demyelination, microglial activation, and immune cell infiltration. 1H10 also remarkably inhibited EAE-associated blood-brain barrier (BBB) disruption, MMP-9 activation, and aberrant synaptic zinc patch formation. Furthermore, the present study showed that long-term treatment with 1H10 also reduced the clinical course of EAE. Therefore, the present study suggests that zinc chelation and AMPK inhibition with 1H10 may have great therapeutic potential for the treatment of multiple sclerosis.

## 1. Introduction

Multiple sclerosis (MS) is a cell-mediated autoimmune disease directed against myelin antigens of the central nervous system (CNS) [1]. The pathological process of MS includes demyelination, multifocal inflammation, blood–brain barrier disruption, encephalitogenic immune cells, reactive gliosis, oligodendrocyte loss, and axonal degeneration [2]. Experimental autoimmune encephalomyelitis (EAE) is a widely used animal model of MS [3], and the progression of EAE occurs in three consecutive steps: The first step is a peripheral immune response (the generation of anti-myelin antibodies and activation of monocytes, T lymphocytes, and B lymphocytes); the second is the disruption of the BBB, followed by the infiltration of encephalitogenic immune cells from the peripheral nervous system (PNS) to the CNS; and the third is the demyelination of nerve fibers [4,5]. The infiltrate is mainly composed of T and B lymphocytes and macrophages [1,6].

The molecular mechanisms that underlie these demyelinating diseases still remain unclear. Our lab recently demonstrated that the pathological disruption of zinc homeostasis during EAE is involved in demyelination and disease pathogenesis. Released zinc can induce the degradation of the extracellular matrix and the matrix metallopeptidase-9 (MMP-9)-dependent breakdown of the BBB, resulting in the migration of encephalitogenic immune cells and demyelinating antibodies. It also activates microglia and induces the release of proinflammatory cytokines, which cause damage to the myelin sheath [7]. Our previous studies also demonstrated that the oral administration of clioquinol (CQ), a well-characterized zinc chelator, or the genetic deletion of zinc transporter 3 (*ZnT3*)—which depletes zinc in synaptic vesicles—decreased the symptoms of and pathological changes in EAE [8,9].

AMP-activated protein kinase (AMPK) is a serine/threonine kinase consisting of an α subunit and regulatory β and γ subunits. It serves as an integrator of energy balance and energy-dependent responses at the cell, tissue, and organism levels to facilitate context-specific responses to changes in the metabolic status [10,11]. AMPK regulates many aspects of cellular metabolism. It is strongly induced by ATP depletion and other related stimuli to restore the cellular energy balance, but its overactivation is deleterious in pathological conditions, such as stroke [12] and neurodegenerative diseases [13,14]. 1H10 is a novel chemical inhibitor of AMPK that was discovered as a potential agent to protect against stroke-related injury [15]. It has been reported that its administration protected against middle cerebral artery occlusion (MCAO)—induced brain injury and zinc-induced neurotoxicity. However, its protective effects have never been tested in MS.

Using MOG_35-55_-induced experimental autoimmune encephalomyelitis (EAE) as a model for MS [3], we investigated the therapeutic potential of 1H10 to protect against disease progression and the pathological changes induced by zinc-mediated pathogenic mechanisms during EAE. Here, we found that 1H10 reduced the severity of EAE and attenuated demyelination, microglial activation, BBB disruption, MMP-9 activation, encephalitogenic immune cell infiltration, and the formation of abnormal zinc patches. Our findings highlight that a new zinc-chelating agent, 1H10, could be a promising therapeutic tool for MS treatment.

## 2. Results

### 2.1. 1H10 Has Zinc-Chelating Capacity

To assess whether a new potential zinc chelator, 1H10 (Figure 1A), actually exhibited such chelating capacity, we performed intracellular zinc staining in mouse neuronal cultures and dose-response zinc-binding assays in test tubes. We observed a time-dependent increase in intracellular free zinc after the exposure of cortical cultures to 300 μM zinc for 15 min, which was markedly attenuated by 1H10 (Figure 1B). Whereas 40 μM 1H10 significantly reduced FluoZin-3 fluorescence (Kd[Zn^2+^] = 15 nM) within 30 min of zinc treatment, 10 or 20 μM 1H10 actually reduced FluoZin-3 fluorescence from 10 min (Figure 1B). To confirm the zinc-binding capacity of 1H10, we used another fluorescence dye for free zinc detection, Newport Green DCF (Kd[Zn^2+^] = 1 μM), in test tubes. Different doses of 1H10 (0.2~100 μM) were incubated with 20 μM ZnCl_2_ in test tubes, compared with clioquinol or ionomycin. Here, we used clioquinol as a reference zinc chelator and ionomycin as a calcium ionophore. We calculated the half-maximal inhibitory concentration (IC_50_), representative of the half-maximal binding concentrations for drugs against 20 μM ZnCl_2_. 1H10 had an IC_50_ of 10.01 μM, which was higher than that of 2.268 µm for clioquinol. However, ionomycin did not affect the free zinc levels at all (Figure 1C). Since the IC_50_ of 1H10 was four times that of clioquinol, a moderate zinc-binding chemical (Kd[Zn^2+^] = 0.07 μM) [16], we suggested that 1H10 was a very weak zinc-binding compound, in addition to being a new chemical inhibitor for AMPK [15].

### 2.2. 1H10 Reduces EAE-Induced MS Symptoms, Demyelination, and Microglial/Macrophage Activation

In order to test whether zinc chelation and AMPK inhibition by 1H10 could ameliorate the clinical signs and progression of the disease in EAE, mice were immunized with MOG_35-55_ peptide to induce EAE, 1H10 was delivered by intraperitoneal injection once per day at a dose of 5 μg/kg/day, and clinical scores were recorded daily until 21 days after EAE induction (Figure 2A). The vehicle-treated mice developed severe EAE symptoms with an incidence rate of 100%. The clinical signs of EAE first appeared on day 11 and peaked on day 18. However, the concurrent administration of 1H10 from the beginning of MOG immunization reduced the clinical scores and incidence rate of induced EAE. 1H10 treatment reduced clinical scores at day 18 from 2.44 ± 0.57 (of 5 maximum) to 0.29 ± 0.18, an 88% reduction (Figure 2B–D). Statistically significant effects on clinical scores were present according to group (F = 11.082, *p* < 0.001), day (F = 14.691, *p* < 0.001), and the interaction between group and day (F = 9.429, *p* < 0.001).

It is well known that the extent of EAE-induced paralysis strongly correlates with the amount of peripheral immune cell infiltration in the white matter of the spinal cord, and this process is known to trigger demyelination and inflammation. We therefore performed immunofluorescence staining to evaluate both demyelination and microglia/macrophage-driven inflammation in the white matter of the spinal cord from vehicle- and 1H10-treated mice after sham surgery or EAE induction. Demyelination was monitored using an antibody against the myelin-specific marker myelin basic protein (MBP), and microglia/macrophage-driven inflammation was monitored using an antibody against F4/80 (Figure 3A). The MBP signal showed that while the sham-operated mice and 1H10-treated EAE mice showed intact white matter in the spinal cord, the vehicle-treated EAE mice had apparent dark regions representing damaged myelin. Thus, 1H10 can inhibit demyelination in EAE mice (Figure 3B). The F4/80 signal consistently showed that the vehicle-treated EAE mice, but not the sham-operated groups or 1H10-treated EAE mice, had obvious microglia/macrophage activation in the white matter of the spinal cord (Figure 3C). Importantly, the regions showing fewer MBP signals were co-located with the F4/80-positive regions in the EAE group, which indicates that 1H10 not only decreases demyelination but also obviously reduces the degree of microglia/macrophage-driven inflammation in EAE mice.

Ionized calcium-binding adaptor molecule 1 (Iba-1) is specifically expressed in microglia and macrophages. Staining against the microglia/macrophage marker, Iba-1, revealed similar patterns of inflammation regions, such as F4/80. Next, we examined the expression of CD68 in Iba-1-positive microglia and macrophages to specifically evaluate their polarization states. CD68, a prominent lysosome-associated membrane protein, is known to be a marker of M1 polarized microglia/macrophages [17] and to functionally regulate phagocytosis [18]. We found that most of Iba-1-positive microglia and macrophages remarkably expressed CD68 at 21 days following EAE induction (Figure 3D). However, 1H10 treatment not only reduced Iba-1 (Figure 3E) and CD68 (Figure 3F) immunoreactivity but also significantly decreased the colocalization coefficient (Figure 3G,H) in EAE mice. To investigate whether 1H10 could also suppresses astrocyte activation after EAE, we performed immunofluorescence staining with an antibody against the astrocyte-specific marker glial fibrillary acidic protein (GFAP) (SI, Appendix A). The vehicle-treated EAE mice displayed a prominent activation of astrocytes and the development of astrogliosis in the white matter of the spinal cord. By contrast, 1H10 treatment significantly reduced astrocyte activation in EAE mice (SI, Appendix A). These findings indicate that 1H10 treatment protects the structure of the myelin sheath by inhibiting the demyelination and inflammation that occurs in the spinal cord of EAE mice and—together with our results showing that it can reduce the severity of EAE—demonstrate that this AMPK inhibitor, with its accompanying zinc-chelating ability, dramatically reduces EAE-induced disease symptoms.

### 2.3. Suppression of EAE-Induced Aberrant Zinc Patches, MMP-9 Activation, and BBB Breakdown in 1H10-Treated Mice

Previously, it was observed that the formation of zinc patches in the spinal cord of EAE mice and these pathological zinc liberations caused EAE-induced white matter destruction and motor deficits [8]. Therefore, to test whether 1H10 treatment could suppress the formation of EAE-induced aberrant zinc-patches, we performed *N*-(6-methoxy-8-quinolyl)-para-toluenesulfonamide (TSQ) staining at 21 days following the initial immunization with MOG_35-55_. TSQ fluorescence signals represent chelatable zinc in the spinal cord. Patch-like TSQ fluorescence, which represents abnormal zinc accumulation, was intense in the spinal cord of EAE mice. However, 1H10-treated EAE mice showed a substantial reduction in abnormal zinc patches in the spinal cord (Figure 4A,B). We also assessed whether the 1H10 treatment could reduce the high content of synaptic-vesicular zinc, which is attributable to the formation of aberrant zinc patches, during EAE. The accumulation of synaptophysin, one of the membrane proteins in presynaptic vesicles, is known to be a reliable marker of axonal damage in the central nervous system (CNS) under inflammatory/demyelinating conditions [19]. In the present study, we found that *ZnT3* immunoreactivity was colocalized with synaptophysin immunoreactivity. Interestingly, the vehicle-treated EAE mice group showed an increased expression of synaptophysin and *ZnT3* in the spinal cord while the 1H10-treated EAE mice group showed significantly reduced synaptophysin and *ZnT3* immunoreactivity (Figure 4C–E). Pathological zinc liberation causes a series of events, such as MMP-9 activation and BBB disruption, that result in increasingly deleterious effects. We examined whether 1H10 treatment influenced EAE-induced MMP-9 activation. MMP-9 activity was significantly increased in the spinal cord white matter of EAE mice. However, 1H10 treatment significantly reduced this (Figure 4F,G). Next, to evaluate whether 1H10 treatment affected EAE-induced BBB disruption, we checked for the leakage of serum immunoglobulin G (IgG), which is used to assess putative damage to the BBB. The examination of spinal cord sections revealed an obvious extravasation of IgG in EAE mice that was not present in either vehicle- or 1H10-treated mice after sham surgery. Compared to vehicle-treated EAE mice, there was a significant reduction in IgG extravasation in the spinal cord of 1H10-treated EAE mice (Figure 4H–J). We also sought to determine endogenous serum IgG leakage from spinal cord vessels in EAE mice. Sections from EAE mice were stained for endogenous IgG and the endothelial cell marker (CD31) to highlight vascular permeability around spinal cord vessels. The vehicle-treated EAE mice showed pronounced and diffuse IgG immunoreactivity in the vessels of the spinal cord, reflecting endogenous serum protein extravasation, which obscured the boundaries between the vessel segments. By contrast, 1H10 treatment significantly decreased IgG immunoreactivity in vessels, a result suggesting that the 1H10 treatment of EAE mice preserved the integrity of the BBB (Figure 4K).

### 2.4. 1H10 Decreases Immune Cell Infiltration and the Phosphorylation of AMPK in Infiltrated CD8^+^ T Cells Following EAE Induction

Exposure to MOG_35-55_ antigen, a peptide and component of myelin, activates the immune system to induce a subsequent immune response specifically targeting myelin through BBB breakdown and mononuclear cell infiltration from the periphery into the CNS. Thus, we first examined EAE-induced mononuclear cell infiltration. EAE mice displayed an infiltration of numerous mononuclear cells into the white matter of spinal cord. As shown in Figure 5A,B, the number of infiltrated mononuclear cells from mice treated with 1H10 was significantly reduced in the spinal cord white matter. We further investigated the presence of major pathogenic immune cells, such as T and B lymphocytes, considered the main drivers in the pathogenesis of EAE, to determine the mononuclear cell population that infiltrated into the CNS. MOG_35-55_-immunized mice revealed a several-fold increased immunoreactivity of CD4^+^ T, CD8^+^ T, and CD20^+^ B lymphocytes compared to the sham group. The immunoreactivity of CD4^+^ T, CD8^+^ T, and CD20^+^ B lymphocytes that infiltrated into the spinal cord white matter was significantly reduced in 1H10-treated EAE mice (Figure 5C–E). Next, we examined the extent of AMPK phosphorylation in CD8^+^ T lymphocytes that infiltrated into the white matter of the spinal cord following EAE. We found that most of these had drastically increased AMPK phosphorylation. However, 1H10 treatment not only reduced CD8^+^ T lymphocyte and p-AMPK immunoreactivity but also significantly decreased the colocalization coefficient (Figure 5F–H). These experiments showed that 1H10 treatment blocks the infiltration of encephalitogenic immune cells into the white matter of the spinal cord.

### 2.5. Long-Term Protective Effects of 1H10 Following EAE Induction

To assess whether long-term treatment with 1H10 is also effective against disease progression in EAE, 1H10 was given by intraperitoneal injection once per day and clinical scores were recorded daily until 45 days following the initial MOG_35-55_ immunization (Figure 6A). As in the short-term treatment experiments lasting 21 days, long-term treatment with 1H10 decreased the clinical scores and incidence rate. The clinical signs of EAE first appeared on day 13 and peaked on day 22. However, the concurrent administration of 1H10 reduced the clinical scores and incidence rate. 1H10 treatment reduced the clinical score at 22 days from 2.19 ± 0.35 (maximum, 5) to 0.17 ± 0.17, a 92% reduction. Statistically significant effects on clinical scores were present according to group (F = 43.189, *p* < 0.001), day (F = 27.293, *p* < 0.001), and the interaction between group and day (F = 16.330, *p* < 0.001). Moreover, the beneficial effects of 1H10 on the clinical symptoms of EAE were obvious until at least day 45 (Figure 6B–D).

## 3. Discussion

Using MOG_35-55_-induced EAE mice and a new zinc chelator, 1H10, we intended to validate the previously reported role of zinc in EAE disease progression. In the present study, we found a reduction in the clinical signs, disease progression, and pathological changes, such as demyelination, microglia/macrophage activation, BBB disruption, MMP-9 activation, encephalitogenic immune cell infiltration, and aberrant zinc patch formation in the spinal cord white matter, in 1H10-treated mice. 1H10 treatment also reduced AMPK phosphorylation in CD8^+^ T lymphocytes that infiltrated into the spinal cords of EAE mice. Thus, the effect of 1H10 in blocking EAE disease progression occurs mainly through zinc chelation and AMPK inhibition.

It has been increasingly recognized that zinc homeostasis has a major impact on the pathophysiological processes of MS, although the precise mechanism is not known. Zinc is one of the most abundant trace elements, essential for proper CNS function [20]. However, the zinc synaptically released into the extracellular space can reach toxic levels during pathological conditions, such as seizures [21], ischemia [22], traumatic brain injury [23,24], hypoglycemia [25], and multiple sclerosis [7,8,9]. The cytoplasmic influx of synaptically released zinc stimulates the activation of NADPH oxidases, mainly via protein kinase C (PKC) [26]. Since there are zinc finger structures that are critical for the enzymatic function of PKC, zinc can regulate its activity [27]. It has been reported that zinc induces the PKC-dependent activation of NADPH oxidase in cortical neurons and astrocytes, by Noh and Koh et al. [28], and reactive oxygen species (ROS) production from NADPH oxidase leads to DNA damage and poly(ADP-ribose) polymerase-1 (PARP-1) activation in the nucleus, followed by neuronal death [29,30]. Our previous studies sought to evaluate whether vesicular zinc is an important player in EAE-induced damage to the spinal cord white matter. We provided evidence that EAE induces vesicular zinc release from the synaptic terminals and increases the formation of aberrant zinc patches in the spinal cord, which is prevented by the administration of a zinc chelator [8] or NADPH oxidase inhibitor [7]. In addition, *ZnT3* gene deletion, which specifically depletes vesicular zinc in the CNS, also leads to a reduction in the EAE-induced formation of aberrant zinc patches and demyelination in the white matter of the spinal cord [9]. These results strongly suggest that white matter pathology following EAE is induced by a specific sequence of events, such as the liberation of vesicular zinc, followed by subsequent generation of NADPH oxidase derived-ROS, and the generation of a proinflammatory feedback loop. Here, we also found that EAE-induced aberrant zinc patch formation and demyelination were decreased by 1H10 administration. Our findings suggest that the zinc chelation by 1H10 showed protective effects against EAE-induced myelin sheet degeneration.

BBB permeability is affected by highly specialized complexes, such as neurons, pericytes, astrocytic foot processes, and the extracellular matrix (ECM) [31]. The breakdown of the BBB is known to occur in a murine EAE model of MS [32]. Additionally, zinc is a key contributing factor for the activation of MMP-9, a class of zinc-dependent endopeptidases that can degrade the ECM. MMP-9 plays a role in MS, implying the possibility that its activity may regulate the migration of encephalitogenic immune cells through the subendothelial basement membrane. Moreover, MMP-9 can cause demyelination through its proteolytic activity against MBP [33]. Thus, the zinc released from synaptic terminals or damaged neuronal tissue can activate MMP-9, thereby causing BBB disruption and demyelination in the spinal cord of EAE mice. Persistent BBB permeability contributes to progressive demyelination by allowing infiltrating encephalitogenic immune cells or circulating factors, such as fibrinogen, to cross the BBB and attack antigens present on the myelin of the CNS [34]. Furthermore, failed remyelination may be a consequence of persistent BBB permeability-mediated astrogliosis—preventing oligodendrocyte precursors (OPCs) from accessing the destroyed myelin—rather than an actual deficiency of OPCs, in late-stage lesions [35,36]. The present study found that 1H10 treatment reduced EAE-induced pathological outcomes, such as MMP-9 activation, BBB breakdown, encephalitogenic immune cell infiltration, astrogliosis, and myelin sheath damage.

It has been reported that zinc affects the function of immune cells, such as microglia, macrophages, and T and B lymphocytes [37,38,39]. Our previous studies have demonstrated that zinc is also linked to T cell-mediated autoimmune diseases and that the release of endogenous zinc is an upstream event triggering microglial activation. Zinc chelation by CQ attenuated microglial activation and encephalitogenic immune cell infiltration, suggesting that zinc may also be involved in encephalitogenic immune cell-mediated disease progression [8,9]. The present study shows that 1H10 administration also reduced the activation of M1 microglia, mediators of proinflammatory responses, in the spinal cord of EAE mice. Therefore, 1H10 may be able to prevent further inflammatory processes within the spinal cord by silencing microglial cells, reducing the exposure of myelin antigens, infiltration of T lymphocytes, release of proinflammatory cytokines, and recruitment of other encephalitogenic immune cells.

In a murine EAE model of MS, autoreactive T lymphocytes recognize their target autoantigens—such as MBP, MOG, or proteolipid protein—as peptide fragments presented by major histocompatibility complex (MHC) molecules, become activated, and move to the CNS parenchyma, thereby causing myelin sheath destruction. The activated cytotoxic T lymphocytes produce high levels of proinflammatory cytokines, such as tumor necrosis factor-alpha (TNFα) and interferon-gamma (IFNγ). It has been reported that cytotoxic CD8^+^ T lymphocytes reactive to MBP have been shown to induce EAE [40,41,42]. CD8^+^ T lymphocytes can directly recognize and kill antigen-expressing cell types. Activated CD8^+^ T lymphocytes can also mediate the killing of target cells by the Fas-Fas ligand (FasL) pathway, or by the release of cytotoxic granules at the effector/target cell junction [43,44,45]. Moreover, AMPK is well known to be a fundamental regulator of T cell metabolism, preserving cellular energy homeostasis [46]. AMPK is activated in CD8^+^ T lymphocytes in response to energy stress, such as during infection and inflammation [47,48]. AMPK activation is required for the survival of CD8^+^ T lymphocytes during infection and in the tumor microenvironment [49]. In light of this evidence, we confirmed the extent of AMPK phosphorylation with vehicle and 1H10 treatment in MOG_35-55_-induced EAE mice. The present study found that 1H10 administration diminished the infiltration of CD4^+^ and CD8^+^ T and CD20^+^ B lymphocytes in the spinal cords of the mice. In addition, the CD8^+^ T lymphocytes that infiltrated into the spinal cord white matter had significantly increased AMPK phosphorylation in the EAE mice, which was prevented by administration of 1H10. The effect of 1H10 could also be due to a reduced activation of the immune cells in the periphery during the EAE induction and it could explain why the EAE severity, the BBB disruption, and the immune cell infiltration are decreased.

Altogether, this study demonstrates that vesicular zinc and AMPK activation is involved in several steps of MS pathogenesis. The attenuation of EAE’s severity by 1H10 suggests that zinc chelation and AMPK inhibition have great therapeutic potential for treating multiple sclerosis.

## 4. Materials and Methods

### 4.1. Animals

C57BL/6 female mice, aged 8 weeks (18–22 g), were purchased from Daehan Biolink (DBL, Chungcheongbuk, Korea). Mice were housed in a temperature- and humidity-controlled environment (22 ± 2 °C and 55% ± 5% relative humidity under a 12 h light/12 h dark cycle) and supplied with the Purina diet (Purina, Gyeonggi, Korea) and water ad libitum. Animal use and relevant experimental procedures were approved by the Institutional Animal Care and Use Committee, Hallym University (Protocol # Hallym 2014-89; Date of approval: February 11, 2015). This study was written up in accordance with the ARRIVE (Animal Research: Reporting In Vivo Experiments) guidelines [50].

### 4.2. EAE Induction and Clinical Evaluation

EAE was induced as previously described [7,8,9]. Briefly, mice were immunized by the subcutaneous injection of 200 μL of a mixture of recombinant myelin oligodendrocyte glycoprotein 35-55 (MOG_35-55_, AnaSpec, Fremont, CA, USA) in a mixture of incomplete Freund’s adjuvant (IFA, Sigma-Aldrich, St. Louis, MO, USA) and *Mycobacterium tuberculosis* H37RaA (Difco Laboratories, Detroit, MI, USA). Pertussis toxin (PT, List Biological Laboratories, Campbell, CA, USA) was intraperitoneally injected at a dose of 400 ng on post-immunization days 0 and 2. A booster injection of MOG_35-55_ was given on day 7. The clinical signs of EAE were followed and scored daily on a 0–5 scale, on which 0 = no deficit; 0.5 = partial loss of tail tone or slightly abnormal gait; 1.0 = complete tail paralysis or both a partial loss of tail tone and mild hind limb weakness; 1.5 = complete tail paralysis and mild hind limb weakness; 2.0 = tail paralysis with moderate hind limb weakness (evidenced by frequent foot dropping between the bars of the cage top while walking); 2.5 = no weight-bearing on hind limbs (dragging) but with some leg movement; 3.0 = complete hind limb paralysis with no residual movement; 3.5 = hind limb paralysis with mild weakness in forelimbs; 4.0 = complete quadriplegia but with some movement of head; 4.5 = moribund; and 5.0 = dead.

### 4.3. 1H10 Administration and Experimental Design

1H10 was dissolved in DMSO and diluted with saline. 1H10 was intraperitoneally injected once per day at a dose of 5 ug/kg for the entire experimental course. Mice were divided into four groups for histological evaluation on day 21 post-immunization: (1) Sham without 1H10 (vehicle only, *n* = 5), (2) sham with 1H10 (1H10 only, *n* = 5), (3) EAE without 1H10 (EAE + Vehicle, *n* = 8), and (4) EAE with 1H10 (EAE + 1H10, *n* = 7). They were also divided into four groups to evaluate EAE-induced motor deficits on day 45: (1) Sham without 1H10 (vehicle only, *n* = 5), (2) sham with 1H10 (1H10 only, *n* = 5), (3) EAE without 1H10 (EAE + Vehicle, *n* = 8), and (4) EAE with 1H10 (EAE+1H10, *n* = 6).

### 4.4. Tissue Preparation and Cresyl Violet Staining

Mice were deeply anesthetized with urethane (1.5 g/kg, intraperitoneal) in sterile 0.9% NaCl at a volume of 0.01 mL/g body weight. Toe pinch was used to evaluate the effectiveness of the anesthesia. The mice were transcardially perfused with 0.9% NaCl and then with 4% paraformaldehyde (PFA) in phosphate-buffered saline (PBS). The spinal cords were post-fixed with 4% PFA in PBS for 1 h and then immersed in 30% sucrose for cryo-protection. Thereafter, the spinal cord was frozen and coronally sectioned with a cryostat microtome at a 30-μm thickness. To evaluate the infiltration of mononuclear cells, the thoracic spinal cord sections were stained with cresyl violet. For the quantification of the cresyl violet staining, five sections from each mouse were examined from five mice per group. These sections were coded and given to a blinded experimenter, who manually counted the numbers of infiltrating mononuclear cells in the spinal cords. Data were expressed as the average numbers of cresyl violet-positive cells.

### 4.5. Immunohistochemistry

To block endogenous peroxidase activity, sections were immersed in 1.2% hydrogen peroxide for 15 min at room temperature. After washing in PBS, the sections were incubated with primary antibodies in PBS containing 0.3% Triton X-100 at 4 °C overnight as follows: Monoclonal rat anti-CD4 (diluted 1:50, BD Bioscience, San Jose, CA, USA) and CD8 (diluted 1:50, BD Bioscience, San Jose, CA, USA) antibodies or a polyclonal goat anti-CD20 antibody (diluted 1:50, SantaCruz Biotechnology, Dallas, TX, USA). After washing in PBS, the sections were incubated in biotinylated anti-rat IgG (diluted 1:250; Vector, Burlingame, CA, USA) to detect CD4 and CD8 antibody, biotinylated anti-goat IgG (diluted 1:250; Vector, Burlingame, CA, USA) to detect CD20 antibody, or biotinylated anti-mouse IgG to detect endogenous IgG, for 2 h at room temperature. Thereafter, sections were immersed in avidin-biotin-peroxidase complex (Vector, Burlingame, CA, USA) for 2 h at room temperature. Between incubations, the sections were washed with PBS. The immune reaction was visualized with 3,3′-diaminobenzidine (Sigma-Aldrich, St. Louis, MO, USA) in 0.01 M PBS containing 0.015% H_2_O_2_, and the sections were mounted on gelatin-coated slides and coverslipped with Canada Balsam (Junsei chemical Co., Ltd., Chuo-ku, Tokyo, Japan). The immunoreactions were observed under an Olympus IX70 inverted microscope. To evaluate nonspecific effects, a few sections in every experiment were incubated in a buffer without any primary antibodies. This procedure always resulted in a complete lack of immunoreactivity. To quantify CD4, CD8, CD20, and IgG immunoreactivity, five coronal sections were analyzed by a blinded experimenter using ImageJ (National Institute of Health, Bethesda, Rockville, MD, USA). The immunofluorescence intensity and area of CD4, CD8, CD20, and IgG were expressed as the mean gray value and % area, respectively.

### 4.6. Immunofluorescence Analysis

Immunofluorescence labeling was performed as per routine immunostaining protocols, such as those referenced above. The primary antibodies used in this study were as follows: Rabbit anti-myelin basic protein (MBP; diluted 1:500; Invitrogen, Carlsbad, CA, USA); goat anti-Iba-1 (Iba-1; diluted 1:500; Abcam, Cambridge, UK); rat anti-CD68 (diluted 1:100; Bio-Rad Laboratories, Hercules, CA, USA); rat anti-CD8 (diluted 1:50; BD Bioscience San Jose, USA); rat anti-F4/80 (diluted 1:100; eBioscience, San Diego, CA, USA); rabbit anti-phospho-AMPKα 1/2 (diluted 1:200; Abcam); mouse anti-CD31 (diluted 1:200; Millipore, Cambridge, UK); rabbit anti-MMP-9 (diluted 1:200; Abcam, Cambridge, UK); rabbit anti-GFAP (diluted 1:500; Abcam, Cambridge, UK); mouse anti-synaptophysin (diluted 1:200; Cell Signaling Technology, Danvers, MA, USA); and rabbit anti-*ZnT3* (diluted 1:200; Synaptic Systems, Göttingen, Germany). For double labeling, primary antibodies were simultaneously incubated and further processed for each antibody. For visualization of the primary antibody binding, fluorescent-conjugated secondary antibodies were applied: Alexa 488 and 594 (diluted 1:250; Invitrogen, Carlsbad, CA, USA). Sections were counterstained with DAPI (4,6-diamidino-2-phenylindole; diluted 1:1000; Invitrogen, Carlsbad, CA, USA). Fluorescence-stained sections were mounted on gelatin-coated slides and coverslipped with dibutylphthalate polystyrene xylene (DPX, Sigma-Aldrich, St. Louis, MO, USA). Fluorescence signals were detected using a Zeiss LSM 710 confocal microscope (Carl Zeiss, Oberkochen, Germany) with a sequential scanning mode for DAPI and Alexa 488 and 594. Stacks of images (1024 × 1024 pixels) from consecutive slices of 0.5–0.8 μm in thickness were obtained by averaging 15 scans per slice and were processed using ZEN 2 (blue edition, Carl Zeiss, Oberkochen, Germany). Images were taken from the thoracic spinal cord section. The quantification of the mean intensity and colocalization experiments was performed using the ZEN 2 software (blue edition, Carl Zeiss, Oberkochen, Germany). The overlap coefficient (Manders’ coefficient) was used as the colocalization coefficient. The area of immunoreactivity was measured using ImageJ (National Institute of Health, Bethesda, Rockville, MD, USA) and expressed as % area. In addition, five sections from each mouse were scored by a blinded experimenter to quantify microglial activation. The criteria for microglial activation were based on the number of F4/80 immunoreactive cells and their morphology [37].

### 4.7. Zinc Staining (TSQ Method)

The *N*-(6-methoxy-8-quinolyl)-para-toluenesulfonamide (TSQ) histochemical method was used as previously described [51,52]. Briefly, mice were sacrificed on day 21 post-immunization by decapitation under 5% of isoflurane anesthesia, and the brains were removed and frozen in powdered dry ice. The frozen unfixed spinal cords were coronally sectioned at a 20-μm thickness in a −15 °C cryostat and then thawed on gelatin-coated slides and air-dried. The sections were immersed in a solution of 4.5 μm TSQ (Molecular Probes, Eugene, OR, USA) in 140 mM sodium barbital and 140 mM sodium acetate buffer (pH 10.5–11) for 60 s, and then rinsed for 60 s in 0.9% saline. TSQ binding was imaged with a fluorescence microscope (Olympus upright microscope, epi-illuminated with 360 nm UV light) and photographed through a 500-nm long-pass filter using an INFINITY3-1 CCD cooled digital color camera (Lumenera Co., Ottawa, ON, Canada) with the INFINITY Analyze software (the release version 6.0). The intensity of TSQ was measured using ImageJ (NIH, Bethesda, MD, USA) and expressed as mean gray values.

### 4.8. Cell Culture

Both neuronal and mixed glial neuronal cultures were prepared from embryonic mice at 13–14 days as described previously [53]. In brief, the growth medium consisted of Dulbecco’s Modified Eagle Medium (DMEM, GibcoBRL, Grand Island, NY, USA) with 2 mM glutamine, 5% fetal bovine serum, and 5% horse serum. Minced cerebral cortices combined with growth medium were seeded onto a poly-D-lysine (Sigma, St. Louis, MO, USA) pre-coated plate at 8–9 hemispheres per 24-well plate. The cultures were incubated at 37 °C in a humidified 5% CO_2_ atmosphere. All cultures were used at 10–14 days in vitro. These experiments were performed under the guidelines for the care and use of mice in research and under protocols approved by the Animal Care and Use Committee of Sejong University.

### 4.9. Detection of Zinc-Chelating Capacity in Cell Cultures or in Test Tubes

To detect the amount of intracellular free zinc in mouse neuronal cultures, we pre-loaded 5 μM FluoZin-3 (Kd[Zn^2+^] = 15 nM, Molecular Probes, Eugene, OR, USA) in Eagle’s Minimal Essential Medium (MEM, GibcoBRL) for 30 min. Then, 300 μM ZnCl_2_ in Hank’s balanced salt solution (HBSS, Biowest, MO, USA; supplemented with 1.8 mM CaCl_2_, 1.22 μM MgSO_4_, 3.15 μM MgCl_2_, and 1.94 mM glucose) was treated to cortical cultures for 15 min. Subsequently, the cortical cultures were washed with HBSS to eliminate extracellular zinc, and then 1H10 (10, 20, or 40 μM) was used for the post-treatment of the cultures. The relative fluorescence units (RFUs) of FluoZin-3 were measured with a fluorescence microplate reader at Ex/Em = 494/516 nm (Molecular Devices, Sunnyvale, CA, USA) at 10-min intervals for up to 1 h. The capacity for zinc chelation was measured using Newport Green DCF (dipotassium salt, Kd[Zn^2+^] = 1 μM, Molecular Probes) in test tubes. For this, 0.1 μM Newport Green DCF and 20 μM zinc chloride were mixed—with or without the varied concentrations of 1H10, clioquinol (positive control) or ionomycin (negative control)—in HEPES-buffered saline (135 mM NaCl, 5 mM KCl, 1.5 mM CaCl_2_, 10 mM HEPES (pH 7.4), 6 mM glucose). The RFUs were measured with a fluorescence microplate reader at Ex/Em = 505/535 nm (Molecular Devices) after 30 min. Log–phase graphs showing the dose-dependent zinc-chelating capacity and half-maximal inhibitory concentration (IC_50_) values were calculated by using a commercial scientific 2-D graphing and statistics program (Prism 5; GraphPad Software, La Jolla, CA, USA).

### 4.10. Statistical Analysis

All data were reported as mean ± SEM. Repeated measure ANOVAs were conducted to investigate differences in the clinical scores over time among groups using SPSS ver.21. Other comparisons between vehicle- and 1H10-treated groups were performed with a two-tailed unpaired Student’s *t*-test. In order to compare the values among 4 groups, the remaining data were analyzed by the Kruskal-Wallis test with post-hoc analysis using Bonferroni correction. *p*-values less than 0.05 (*p* < 0.05) were considered to be statistically significant.

## Figures and Tables

**Figure 1 ijms-21-03375-f001:**
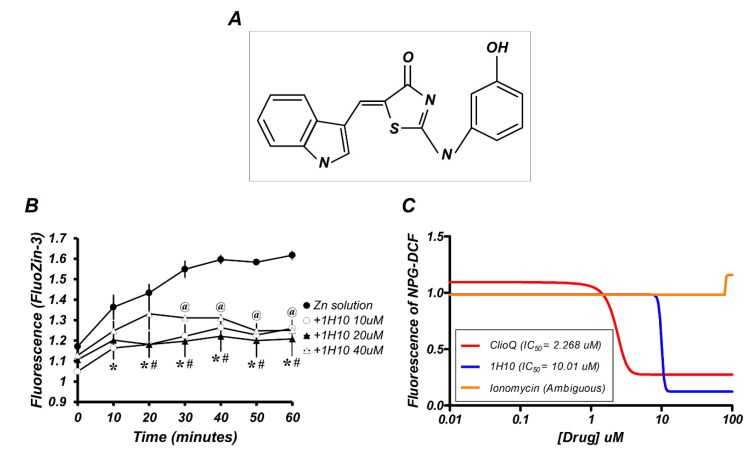
1H10 has a zinc-chelating capacity. (**A**) Chemical structure of 1H10. (**B**) FluoZin-3 fluorescence level was measured at the indicated time points after 15 min of exposure to 300 µm zinc with or without the three different concentrations of 1H10 (10, 20, or 40 µM) in mouse neuronal cultures. Data are mean ± SEM (*n* = 3). @*,* #*,* and *: *p* < 0.05 versus zinc alone at the same time point (unpaired Student’s *t*-test). (**C**) In the test tubes, the fluorescence of Newport Green DCF (NPG-DCF, 0.1 μM) was estimated after a 30-min incubation of zinc (20 µM) with or without the drugs (ClioQ; clioquinol, 1H10, or ionomycin) at the different doses (0, 0.2, 0.6, 2, 10, 20, 40, or 100 µM). The half maximal inhibitory concentration (IC_50_) was calculated by Prism 5.

**Figure 2 ijms-21-03375-f002:**
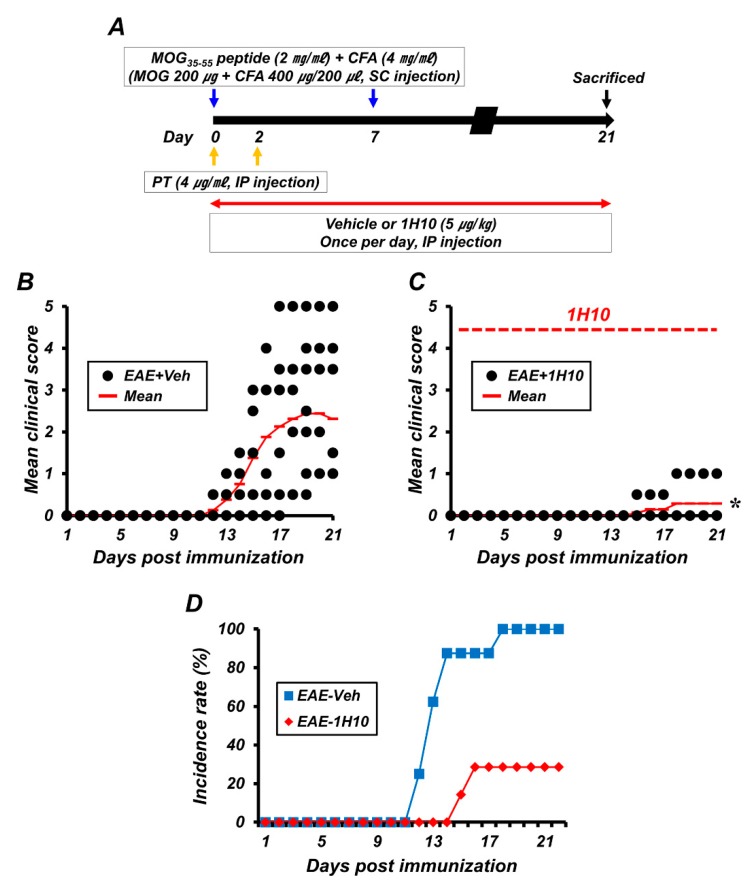
1H10 ameliorates the clinical signs and progression of myelin oligodendrocyte glycoprotein 35-55 (MOG_35-55_)-induced experimental autoimmune encephalomyelitis (EAE). (**A**) Timeline showing the experimental design. 1H10 was intraperitoneally administered once per day for the entire experimental course. Mice were then killed on day 21 following the initial immunization. (**B**,**C**) EAE clinical scores are shown for both vehicle (**B**) and 1H10 groups (**C**) (mean ± SEM; n = 7–8 per group). @, #, and *: *p* < 0.05. vehicle-treated EAE mice (repeated measure ANOVA; Day: F = 14.691, *p* < 0.001; Group: F = 11.082, *p* = 0.001; Day × group interaction: F = 9.429, *p* < 0.001). (**B**) Percentage disease incidence of vehicle- or 1H10-treated immunized mice (mean ± SEM; n = 7–8 per group).

**Figure 3 ijms-21-03375-f003:**
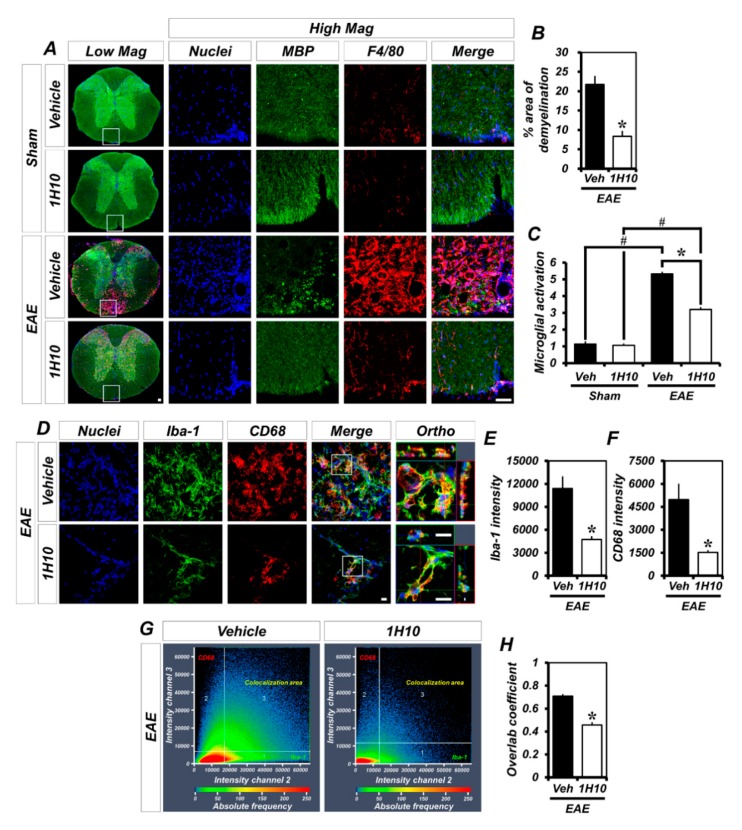
EAE-induced demyelination and microglia/macrophage activation are reduced by treatment with 1H10. (**A**) Representative microglia/macrophage activation in the spinal cord of sham-operated and MOG_35-55_-immunized mice (either vehicle or 1H10) at day 21 as shown by immunofluorescence for F4/80 (red). Demyelinated area shown by reduced myelin basic protein (MBP) staining (green). Nuclei stained with 4′,6-diamidino-2-phenylindole (DAPI) (blue). Scale bar, 50 µm. (**B**,**C**) Bar graphs showing the percent areas of demyelination (**B**) and the grades of microglia/macrophage activation (**C**) as determined in the same spinal cord region (mean ± SEM; *n* = 3–5 per group). * *p* < 0.05 vs. vehicle-treated EAE mice; # *p* < 0.05 vs. sham-operated mice (**B**) unpaired Student’s *t*-test; **C**: Kruskal-Wallis test followed by Bonferroni post-hoc test: Chi square = 13.613, df = 3, *p* = 0.003). (**D**) Representative images of ionized calcium binding adaptor molecule 1 (Iba-1, green) and cluster of differentiation 68 (CD68, red) immunopositive cells as merged images for vehicle- and 1H10-treated EAE mice. Nuclei stained with DAPI (blue). Scale bar, 50 µm. (**E**,**F**) Quantification of the immunofluorescence intensity of Iba-1 (**E**) and CD68 (**F**) as determined in the same spinal cord region (mean ± SEM; *n* = 4 per group). * *p* < 0.05 vs. vehicle-treated EAE mice (unpaired Student’s *t*-test). (**G**) Colocalization scatterplots of Iba-1 with CD68. (**H**) Quantitative colocalization parameters of Iba-1 with CD68 determined by measuring Mander’s overlap coefficient (mean ± SEM; *n* = 4 per group). * *p* < 0.05 vs. vehicle-treated EAE mice (unpaired Student’s *t*-test).

**Figure 4 ijms-21-03375-f004:**
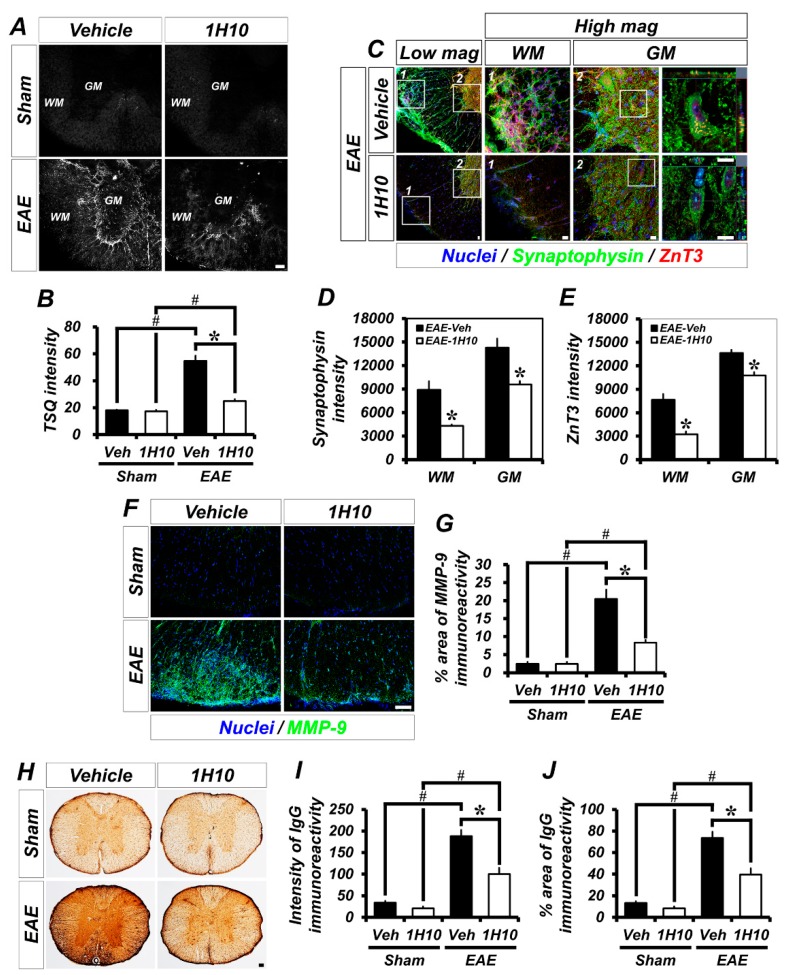
1H10 reduces aberrant synaptic zinc patches, Matrix metallopeptidase 9 (MMP-9) activation, and blood-brain barrier (BBB) disruption in the white matter of the spinal cord in EAE. (**A**) Representative images showing sections of the spinal cord stained with *N*-(6-methoxy-8-quinolyl)-para-toluenesulfonamide (TSQ) to detect zinc accumulation. Scale bar, 100 µm. WM = white matter, GM = grey matter. (**B**) Quantification of TSQ intensity in the white matter of the spinal cord (mean ± SEM; *n* = 3 per group). * *p* < 0.05 vs. vehicle-treated EAE mice; # *p* < 0.05 vs. sham-operated mice (Kruskal-Wallis test followed by Bonferroni post-hoc test: Chi square = 9.359, df = 3, *p* = 0.025). (**C**) Double label confocal micrographs of synaptophysin (green) and zinc transporter 3 (*ZnT3*, red) in the spinal cord from vehicle- and 1H10-treated EAE mice. Scale bar, 20 µm. WM = white matter, GM = grey matter. (**D**,**E**) Quantification of the immunofluorescence intensity of synaptophysin (**C**) and *ZnT3* (**D**) as determined in the same spinal cord region (mean ± SEM; *n* = 3 per group). * *p* < 0.05 vs. vehicle-treated EAE mice (unpaired Student’s *t*-test). (**F**) Immunofluorescence images representing the expression of MMP-9 in the white matter of the spinal cord. Scale bar, 50 µm. (**G**) Bar graph showing the percent areas of MMP-9 immunoreactivity in the thoracic spinal cord with or without 1H10 treatment in sham-operated and EAE mice (mean ± SEM; *n* = 3 per group). * *p* < 0.05 vs. vehicle-treated EAE mice; # *p* < 0.05 vs. sham-operated mice (Kruskal-Wallis test followed by Bonferroni post-hoc test: Chi square = 9.392, df = 3, *p* = 0.025). (**H**) Photomicrographs showing sections of the spinal cord stained for anti-mouse immunoglobulin G (IgG) to detect endogenous IgG. Scale bar, 100 µm. (**I,J**) Graph representing the intensity (**I**) or percent area (**J**) of IgG leakage from the spinal cord in mice treated with vehicle or 1H10 at 3 weeks following the initial immunization (mean ± SEM; *n* = 4–5 per group). * *p* < 0.05 vs. vehicle-treated EAE mice; # *p* < 0.05 vs. sham-operated mice (Kruskal-Wallis test followed by Bonferroni post-hoc test: Chi square = 15.436, df = 3, *p* = 0.001). (**K**) Double label confocal micrographs of CD31^+^ endothelial cells (red) and endogenous mouse IgG molecules (green) in the white matter of the spinal cord from vehicle- and 1H10-treated EAE mice. Scale bar, 20 µm.

**Figure 5 ijms-21-03375-f005:**
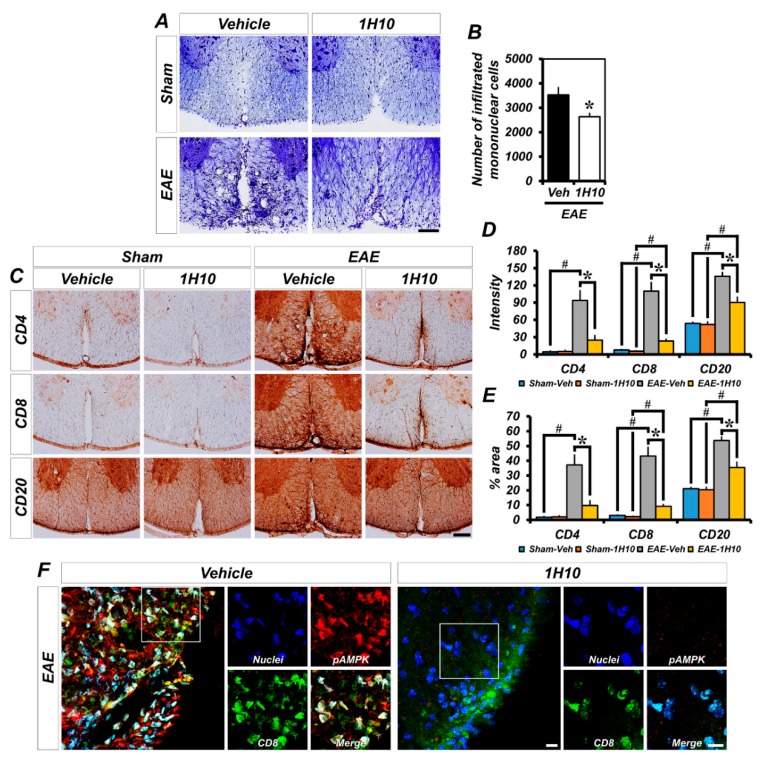
1H10 treatment inhibits EAE-induced immune cell infiltration into the white matter of the spinal cord and the phosphorylation of AMP-activated protein kinase (AMPK) in infiltrated CD8^+^ T cells. (**A**) Representative images showing sections of the spinal cord stained for cresyl violet to detect the infiltration of mononuclear cells. Scale bar, 100 µm. (**B**) Quantification of infiltrated mononuclear cells from the spinal cord in mice treated with vehicle or 1H10 at 3 weeks following initial immunization (mean ± SEM; *n* = 5 per group). * *p* < 0.05 vs. vehicle-treated EAE mice (unpaired Student’s *t*-test). (**C**) Representative images showing the expression of the T and B cells stained with antibodies against cell surface molecules, such as CD4, CD8, and CD20. (**D**,**E**) Bar graph showing the intensity (**D**) or percent area (**E**) of CD4, CD8, and CD20 immunoreactivity in the thoracic spinal cord with or without 1H10 treatment in sham-operated and EAE mice (mean ± SEM; *n* = 4–5 per group). * *p* < 0.05 vs. vehicle-treated EAE mice; # *p* < 0.05 vs. sham-operated mice (Kruskal-Wallis test followed by Bonferroni post-hoc test: CD4: Chi square = 14.895, df = 3, *p* = 0.002; CD8: Chi square = 15.456, df = 3, *p* = 0.001; CD20: Chi square = 14.842, df = 3, *p* = 0.002). (**F**) Representative immunofluorescence images showing CD8^+^ T cells that were co-labeled with the phospho-AMPKα 1/2 in the spinal cord from vehicle- and 1H10-treated EAE mice. (**G**) Colocalization scatterplots of CD8 with phospho-AMPKα 1/2. (**H**) Quantitative colocalization parameters of CD8 with phospho-AMPKα 1/2 determined by measuring Mander’s overlap coefficient (mean ± SEM; *n* = 4 per group). * *p* < 0.05 vs. vehicle-treated EAE mice (unpaired Student’s *t*-test).

**Figure 6 ijms-21-03375-f006:**
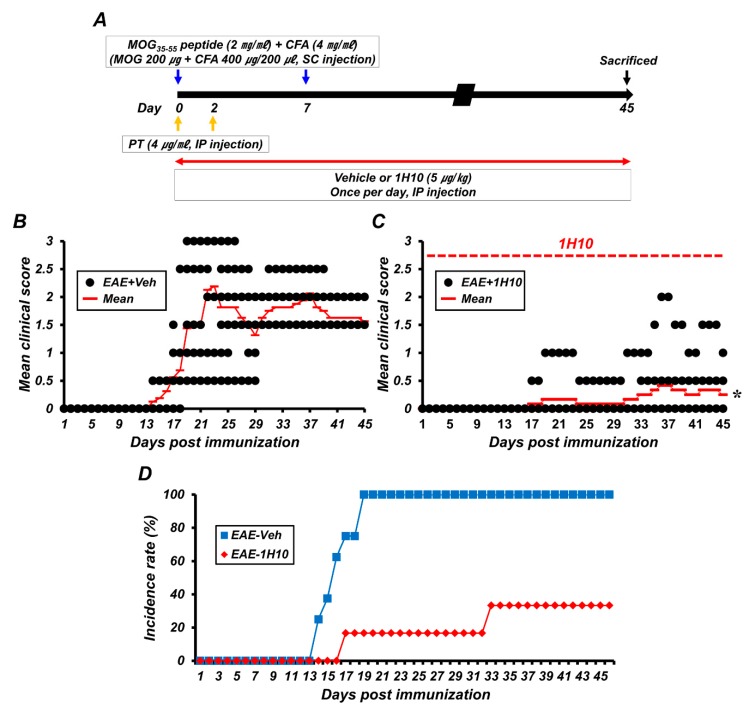
Long-term protective effects of 1H10 after EAE induction. (**A**) Timeline showing the experimental design. 1H10 was intraperitoneally administered once per day during the entire period. Mice were then killed on day 45 after initial immunization. (**B**,**C**) EAE clinical scores are shown for both vehicle (**B**) and 1H10 groups (**C**) (mean ± SEM; n = 6–8 per group). * *p* < 0.05 vs. vehicle-treated EAE mice (repeated measure ANOVA; Day: F = 43.189, *p* < 0.001; Group: F = 27.293, *p* < 0.001; Day × group interaction: F = 16.330, *p* < 0.001). (**D**) Percentage disease incidence of vehicle- or 1H10-treated immunized mice (mean ± SEM; *n* = 6–8 per group).

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
