# Peer review of "A Novel Zinc Chelator, 1H10, Ameliorates Experimental Autoimmune Encephalomyelitis by Modulating Zinc Toxicity and AMPK Activation"

_ijms, 2020, doi:10.3390/ijms21093375_

Round 1

Reviewer 1 Report

Major concern

1. In Figure 3B, why % area of MBP was lower in 1H10-treated that vehicle-treated mice? 3B was not consistent with 3A.

Author Response

  1. In Figure 3B, why % area of MBP was lower in 1H10-treated that vehicle-treated mice? 3B was not consistent with 3A.

<Response: We appreciate this reviewer’s comments and apologizes there was a mistake for wrong label in Figure 3B. We measured % area of demyelination in the present study. Thus, we corrected it in the revised manuscript.>

Reviewer 2 Report

A novel zinc chelator, 1H10, ameliorates experimental autoimmune encephalomyelitis by modulating zinc toxicity and AMPK activation. Bo Young Choi et al.

This is an interesting study of the effect of a zinc chelator, 1H10, on EAE severity. In this manuscript, the authors focus mainly on clinical score, BBB disruption, immune cells infiltration and demyelination.

Minor comments:

-Results 3.2, page 6: Please correct or rephrase this sentence: “In order to test whether zinc chelation and AMPK inhibition by 1H10 could ameliorate the clinical signs and disease progression of EAE, C57BL/6 mice which are highly susceptible to MOG peptide immunization and develop chronic progressive EAE with or without 1H10 treatment were immunized with MOG35-55 peptide to induce EAE.”

-Results 3.2, page 9: Please correct or rephrase these sentences: “Iba-1 detects CNS resident microglia and infiltrating monocyte-derived macrophages. Staining against another microglia/macrophages marker, Iba-1, revealed similar patterns of inflammation regions”. Are you speaking about another marker than Iba-1?

-Figure 4.A shows a zinc accumulation in all 1H10 treated animals (sham and EAE). No zinc accumulation can be seen in EAE animals treated with vehicle. Is that normal?

Major comment:

-The authors should prove that the effect of 1H10 is not due to a reduced activation of the immune cells in periphery during the EAE induction. It could explain why the EAE severity, the BBB disruption and the immune cell infiltration are decreased. A good option could be to treat the EAE mice with 1H10 after the apparition of the first symptoms. Furthermore, it is more clinically relevant to treat after the first symptoms.

Author Response

-Results 3.2, page 6: Please correct or rephrase this sentence: “In order to test whether zinc chelation and AMPK inhibition by 1H10 could ameliorate the clinical signs and disease progression of EAE, C57BL/6 mice which are highly susceptible to MOG peptide immunization and develop chronic progressive EAE with or without 1H10 treatment were immunized with MOG35-55 peptide to induce EAE.”

<Response: We corrected it in the revised manuscript.>

-Results 3.2, page 9: Please correct or rephrase these sentences: “Iba-1 detects CNS resident microglia and infiltrating monocyte-derived macrophages. Staining against another microglia/macrophages marker, Iba-1, revealed similar patterns of inflammation regions”. Are you speaking about another marker than Iba-1?

<Response: In this paragraph, we are referring to Iba-1. Microglia/macrophage-driven inflammation was firstly monitored using an antibody against the F4/80 and then we checked the polarization states of microglia/macrophage using antibodies against the Iba-1 and CD68. We corrected it in the revised manuscript as follow: Ionized calcium-binding adaptor molecule 1 (Iba-1) is specifically expressed in microglia/macrophage. Staining against microglia/macrophages marker, Iba-1, revealed similar patterns of inflammation regions such as F4/80.>

-Figure 4.A shows a zinc accumulation in all 1H10 treated animals (sham and EAE). No zinc accumulation can be seen in EAE animals treated with vehicle. Is that normal?

<Response: We apologized there was a mistake. We found wrong labels related to images in Figure 4A. Thus, we corrected it in the revised manuscript.>

Major comment:

-The authors should prove that the effect of 1H10 is not due to a reduced activation of the immune cells in periphery during the EAE induction. It could explain why the EAE severity, the BBB disruption and the immune cell infiltration are decreased. A good option could be to treat the EAE mice with 1H10 after the apparition of the first symptoms. Furthermore, it is more clinically relevant to treat after the first symptoms.

<Response: We appreciate this reviewer’s comments and agree that checking the effects of 1H10 treatment after the apparition of the first symptoms would be a very good suggestion. However, we plan for it to be part of our future experimental plans following the present work.>

Reviewer 3 Report

The manuscript describes the results obtained with 1H10, an inhibitor of AMPK, against the severity of experimental autoimmune encephalomyelitis (EAE) induced in C57BL/6 mice by active immunization with MOG35-55 emulsified in complete Freund’s adjuvant and two doses of pertussis toxin. The investigators previously described the pathogenic involvement of zinc deposition within the CNS of EAE mice. In the current work, the authors explored the effects of 1H10 as zinc chelator (IC50 = 10 mM) and reported EAE protection under two different experimental approaches (21 days, and long-term: 45-days). The results indicate that the daily, i.p. treatment with 5 mg/kg 1H10 is protective against the disease, reduces demyelination, microglia/macrophage activation, reduces astrogliosis, reduces MMP9 activation, preserves BBB integrity which results in reduced infiltration of nucleated cells (CD4, CD8 T cells and B cells), and reduces phosphorylated AMPK levels, all 21 days after EAE induction. The manuscript is well written, the methodology is described mostly in sufficient detail, and the figures have sufficient quality. The findings are relevant and worth publication, however, there are some minor aspects to consider:

  1. A brief description of 1H10 should be included in the abstract.
  2. The authors indicate in two different sections that the immune cells infiltrated into the parenchyma of the CNS across the BBB from the peripheral nervous system. Are the authors referring to the periphery (systemic circulation), and not the PNS?
  3. In the methods, the authors indicate that besides the clinical scores the rest of the analyses were compared statistically by student t-test. However, some of their analyses compare >2 groups. Shouldn’t ANOVA or non-parametric analysis of the variance be used?
  4. In the results section (lines 286), the authors indicate that Iba-1 detects CNS resident microglia but also infiltrating-derived macrophages. Are authors referring to F4/80? Otherwise, the sentence in lines 286-288 makes little sense.
  5. In the same paragraph (lines 286-301), CD68 should be defined, and it’s M1 marker role and inflammatory function better described.
  6. In figure 4, white matter (WM) and grey matter (GM) should be defined.
  7. The results shown in figure 4 (A), are not clearly described. Looking at the images it appears that the intensity of the staining in higher in 1H10 (sham and EAE) when compared with vehicle-treated (sham and EAE). Also, can this measurement be quantified, as done in other cases? This latter comment also applies to Fig. 4B.
  8. Why did the authors focus on pAMPK in CD8+ T cells only? The role of CD4+ T cells in CNS inflammatory demyelination during EAE has been well described.

Author Response

1. A brief description of 1H10 should be included in the abstract.

<Response: We corrected it in the revised manuscript.>

2. The authors indicate in two different sections that the immune cells infiltrated into the parenchyma of the CNS across the BBB from the peripheral nervous system. Are the authors referring to the periphery (systemic circulation), and not the PNS?

<Response: We appreciate this reviewer’s comments. We are referring to the periphery (system circulation), and not the peripheral nervous system.>

3. In the methods, the authors indicate that besides the clinical scores the rest of the analyses were compared statistically by student t-test. However, some of their analyses compare >2 groups. Shouldn’t ANOVA or non-parametric analysis of the variance be used?

<Response: We appreciate this reviewer’s comments. As this reviewer suggested, we reanalyzed the data by the non-parametric Kruskal Wallis test with post hoc analysis using Bonferroni correction to compare the values among 4 groups. We added this information about the statistical analysis of these data in the figure legend and Materials & Methods part of the revised manuscript.>

4. In the results section (lines 286), the authors indicate that Iba-1 detects CNS resident microglia but also infiltrating-derived macrophages. Are authors referring to F4/80? Otherwise, the sentence in lines 286-288 makes little sense.

<Response: In this paragraph, we are referring to Iba-1. Microglia/macrophage-driven inflammation was firstly monitored using an antibody against the F4/80 and then we checked the polarization states of microglia/macrophage using antibodies against the Iba-1 and CD68. We corrected it in the revised manuscript as follow: Ionized calcium-binding adaptor molecule 1 (Iba-1) is specifically expressed in microglia/macrophage. Staining against microglia/macrophages marker, Iba-1, revealed similar patterns of inflammation regions such as F4/80.>

5. In the same paragraph (lines 286-301), CD68 should be defined, and it’s M1 marker role and inflammatory function better described.

<Response: We corrected it in the revised manuscript.>

6. In figure 4, white matter (WM) and grey matter (GM) should be defined.

<Response: We corrected it in the revised manuscript.>

7. The results shown in figure 4 (A), are not clearly described. Looking at the images it appears that the intensity of the staining in higher in 1H10 (sham and EAE) when compared with vehicle-treated (sham and EAE). Also, can this measurement be quantified, as done in other cases? This latter comment also applies to Fig. 4B.

<Response: We apologized there was a mistake. We found wrong labels related to images in Figure 4A. Thus, we corrected it in the revised manuscript. In addition, we performed quantification of TSQ fluorescence, synaptophysin-, and ZnT3-immunoreactivity as this reviewer’s suggestion. Therefore, we added these results in the revised manuscript.>

8. Why did the authors focus on pAMPK in CD8+ T cells only? The role of CD4+ T cells in CNS inflammatory demyelination during EAE has been well described.

<Response: We appreciate this reviewer’s comments and understand reviewer’s concern. Studies regarding the immunobiology of MS and EAE have largely focused on CD4+ T lymphocytes as the main mediator of pathogenesis. However, numerous studies have reported the pathogenic role for CD8+ T lymphocytes in MS and EAE. CD8+ T lymphocytes represent the predominant T-cell population in CNS lesions of MS patients {Booss, 1983, 6607973} and exhibit oligoclonal expansion at the site of pathology {Babbe, 2000, 10934227}{Jacobsen, 2002, 11872611}. Previous studies have demonstrated that AMPK is activated in CD8+ T lymphocytes in response to energy stress, such as during infection and inflammation {Rolf, 2013, 23310952;Blagih, 2015, 25607458}. In addition, AMPK activation is required for survival of CD8+ T lymphocytes during infection and in tumor microenvironment {Rao, 2015, 25760243}. In view of this evidence, we focused on AMPK phosphorylation in CD8+ T lymphocytes.>

Round 2

Reviewer 2 Report

The authors should add in the discussion that the effect of 1H10 could also be  due to a reduced activation of the immune cells in periphery during the EAE induction and that it could explain why the EAE severity, the BBB disruption and the immune cell infiltration are decreased. 

Author Response

The authors should add in the discussion that the effect of 1H10 could also be  due to a reduced activation of the immune cells in periphery during the EAE induction and that it could explain why the EAE severity, the BBB disruption and the immune cell infiltration are decreased. 

<Response: We appreciate this reviewer's comment. According to the reviewer's advice, we added the following sentence in the discussion part of the revised manuscript. "The effect of 1H10 could also be  due to a reduced activation of the immune cells in periphery during the EAE induction and it could explain why the EAE severity, the BBB disruption and the immune cell infiltration are decreased.">